# Factors Affecting the Quality of Canola Grains and Their Implications for Grain-Based Foods

**DOI:** 10.3390/foods12112219

**Published:** 2023-05-31

**Authors:** Rachid Sabbahi, Khalil Azzaoui, Larbi Rhazi, Alicia Ayerdi-Gotor, Thierry Aussenac, Flore Depeint, Mustapha Taleb, Belkheir Hammouti

**Affiliations:** 1Laboratory of Development and Valorization of Resources in Desert Zones, Higher School of Technology, Ibn Zohr University, Quartier 25 Mars, Laayoune 70000, Morocco; 2Laboratory of Engineering, Electrochemistry, Modeling and Environment, Faculty of Sciences, Sidi Mohamed Ben Abdellah University, Fez 30000, Morocco; k.azzaoui@yahoo.com (K.A.); mustaphataleb62@yahoo.fr (M.T.); 3Institut Polytechnique UniLaSalle, Université d’Artois, ULR 7519, UniLaSalle, 19 rue Pierre Waguet, 60026 Beauvais, France; thierry.aussenac@unilasalle.fr (T.A.); flore.depeint@unilasalle.fr (F.D.); 4Institut Polytechnique UniLaSalle, AGHYLE, UP 2018.C101, UniLaSalle, 19 rue Pierre Waguet, 60026 Beauvais, France; alicia.ayerdi-gotor@unilasalle.fr; 5Laboratory of Applied Chemistry and Environment, Faculty of Sciences, Mohammed First University, Oujda 60000, Morocco; b1.hammouti@ump.ac.ma

**Keywords:** *Brassica napus* L., rapeseed, crop, oil, meal, bakery products, fatty acids, glucosinolates, functional properties, food quality

## Abstract

Canola, *Brassica napus* L., is a major oilseed crop that has various uses in the food, feed, and industrial sectors. It is one of the most widely produced and consumed oilseeds in the world because of its high oil content and favorable fatty acid composition. Canola grains and their derived products, such as canola oil, meal, flour, and bakery products, have a high potential for food applications as they offer various nutritional and functional benefits. However, they are affected by various factors during the production cycle, post-harvest processing, and storage. These factors may compromise their quality and quantity by affecting their chemical composition, physical properties, functional characteristics, and sensory attributes. Therefore, it is important to optimize the production and processing methods of canola grains and their derived products to ensure their safety, stability, and suitability for different food applications. This literature review provides a comprehensive overview of how these factors affect the quality of canola grains and their derived products. The review also suggests future research needs and challenges for enhancing canola quality and its utilization in food.

## 1. Introduction

Canola or oilseed rape, *Brassica napus* L., native to Asia and the Mediterranean, is one of the most important oilseed crops in the world, with a wide range of applications in food, feed, and industrial sectors [1]. Canadian plant breeders identified a cultivar whose defatted meal had both low erucic acid (<2%) and glucosinolate (<30 μmol g^−1^) content [2] and named it canola. This rapeseed is a cool-season broadleaf crop that has winter and spring varieties. Winter canola varieties are planted and established in the fall (usually between August and November), survive the winter and resume growth in the spring. Spring canola varieties are planted in the spring (usually after March) and do not require overwintering [3]. From 2018 to 2020, the major global producers of *B*. *napus* were Canada, China, and India, representing roughly 60% of global production and harvested area [4]. Various components, such as oil, protein, carbohydrates, crude fiber, ash, minerals (e.g., calcium, magnesium, potassium, phosphorus, and sulfur), vitamins (e.g., vitamin E, vitamin K, and vitamin B complex), and moisture content, compose canola seeds [5]. These components contribute to the nutritional quality and functional properties of canola seeds and their products. However, the composition of canola seeds varies depending on the variety, growing conditions, harvesting methods, and storage conditions. The composition of canola seeds affects their quality, nutritional value, and industrial applications [6,7].

The most important component of canola seeds is oil, which accounts for 40–50% of the seed weight, depending on the cultivar [8,9]. The oil’s quality is determined by its fatty acid composition, which affects its oxidative stability, flavor, and nutritional value. Scientists consider canola oil as one of the healthiest oils due to its low level of saturated fatty acids (~7%) and a high level of monounsaturated fatty acids (60–65%), mainly oleic acid (C18:1) (~60%), which have beneficial effects on human health. Canola oil also contains essential polyunsaturated fatty acids (28–32%), such as linoleic acid (C18:2) (~21%) and alpha-linolenic acid (C18:3) (~11%), which act as building blocks for omega-6 and omega-3 fatty acids, respectively [10,11,12]. The ratio of omega-6 to omega-3 in canola oil is about 2:1, which is considered optimal for human health. Canola oil has a range of bioactive compounds, such as tocopherols, phytosterols, phenolic acids, and carotenoids, all of which have anti-inflammatory, antioxidant, and anticancer effects [13,14]. Table 1 shows data on the components of canola seed grown in Western Canada.

In addition to the oil, canola meal, and flour, the by-products of oil extraction contain residual oil and protein that can be used for animal and human nutrition. Canola meal contains up to 50% protein, determined on a dry weight basis [15] and a [16]. The amino acid composition of canola proteins is comparable to that of soybean proteins and meets the Food Agriculture Organization (FAO) and World Health Organization (WHO) requirements for adults (Table 2). However, the digestibility of canola proteins is lower than that of animal and some plant proteins due to the presence of anti-nutritional factors such as glucosinolates (GSL), phytates, tannins, and dietary fiber [17]. Canola meal also has a high content of phosphorus and sulfur, which are essential minerals for animal nutrition [18]. It can be used as a source of protein in animal feed production, contributing to the diversification of raw materials in animal feed [18,19].

Canola flour can be used as a human food ingredient, especially for bakery products. It can enhance the nutritional value, antioxidant capacity, and dietary fiber content of grain-based foods, as well as improve their flavor, texture, and shelf-life [22,23,24]. However, canola flour also has some drawbacks, such as low water solubility, high water absorption capacity, poor emulsifying properties, and a bitter taste [14]. Therefore, some modifications are needed to improve the functionality and acceptability of canola flour.

Canola grain quality depends on various factors, such as genotype, environment, harvesting, storage, and processing [3,25,26]. These factors can influence the chemical composition, physical properties, and functional characteristics of canola seeds, oil, and meal. The quality parameters include oil, protein, chlorophyll, total GSLs, free fatty acids (FFAs), and fatty acid composition [27]. Furthermore, the quality of canola grains has implications for grain-based foods in terms of nutritional value, functional properties, and sensory attributes. Therefore, standard methods and specifications are needed to ensure the quality and safety of canola oil and meal for human and animal consumption.

This literature review aims to cover the critical factors affecting the quality of canola grains and their implications for grain-based foods. The objectives are to: (i) summarize the current knowledge on the quality characteristics and evaluation of canola grains, oil, and meal; (ii) review the effects of genotypic, environmental, agronomic, and post-harvest factors on canola quality; (iii) discuss the applications and markets for canola oil, meal, flour, and bakery products; and (iv) identify the knowledge gaps and research needs for improving canola seed quality and their implications for grain-based foods.

## 2. Factors Affecting the Quality of Canola Grains

The quality of canola grains is a crucial factor that affects their market value and usability for various purposes. Many factors can affect the quality of canola grains, including genetic factors, environmental factors, agronomic practices, and post-harvest factors [28]. Figure 1 outlines the different elements that contribute to canola grain quality and illustrates their relationships and interactions. Examining the impact of these factors is essential for optimizing the production and utilization of this valuable crop.

### 2.1. Genetics and Breeding


*Genetic Diversity and Population Structure of Canola Germplasm*


The genetic diversity and population structure of the canola germplasm reflect its origin, history, adaptation, and improvement. Several studies have assessed the genetic diversity and population structure of canola at global or regional scales using molecular markers. For example, Gyawali et al. [29] studied the genetic diversity and population structure of 169 *B*. *napus* lines from different regions using 84 simple sequence repeat (SSR) markers. They found that genetic diversity varied among regions, with Europe having the highest and Australia and Canada having the lowest, and the lines were divided into two subpopulations based on growth habits. This study suggested that the lines with unique alleles could be useful for breeding programs. In another study, the allelic diversity of 72 *B*. *napus* genotypes covering contemporary germplasms in Australia and China, as well as samples from India, Europe, and Canada, was described using 55 polymorphic SSR markers spanning the complete *B*. *napus* genome [30]. Hierarchical clustering and two-dimensional multidimensional scaling separated a Chinese group (China-1) from a “mixed” group of Australian, Chinese (China-2), European, and Canadian lines. A tiny group from India was clearly unique from all other *B*. *napus* genotypes. Chinese genotypes, particularly those in the China-1 group, have inherited unique alleles from interspecific crossing, primarily with *B*. *rapa*, whereas the China-2 group shares several alleles with Australian genotypes. Other researchers analyzed the genetic diversity and population structure of feral rapeseeds by genotyping 537 individuals in Japan using 30 microsatellite markers [31]. The feral rapeseeds showed moderate genetic diversity and high inbreeding and were divided into eight genetic clusters that did not correspond to geographic regions. In addition, some feral rapeseeds had unknown origins, while others were genetically modified, implying that selection and hybridization influenced feral rapeseeds. These studies demonstrate the importance of understanding the genetic diversity and population structure of canola germplasms for germplasm conservation, utilization, and improvement. However, more studies are needed to explore the genetic diversity and population structure of canola germplasms.


*Genetic Architecture and Molecular Mechanisms of Quality Traits in Canola*


Different canola varieties exhibit variations in oil, protein, chlorophyll, GSLs, fatty acids, seed size and weight, and other traits that influence the quality of canola grains and their derived products [32,33]. These differences may affect the nutritional value, processing suitability, and end-use characteristics of the canola grain [18,34]. Breeding for desirable traits, such as increased oil content, an improved fatty acid profile, and enhanced protein content, can lead to the development of canola cultivars with superior grain quality [35]. For this purpose, breeding strategies and genetic improvement techniques have been employed to enhance canola quality by selecting for traits such as high oil content, low erucic acid, GSLs, and sinapine content, and desirable fatty acid profiles [36,37]. These traits are quantitative and complex, meaning they are influenced by multiple genes and environmental factors, and their genetic architecture is not fully understood. However, recent advances in molecular biology and genomics have shed light on some of the genes and molecular mechanisms involved in the biosynthesis and regulation of these traits in canola.

Oil content is one of the most important quality traits in canola, as it determines the yield and quality of canola oil. It is highly heritable and varies from 30% to 50% among canola varieties [38]. Several studies have identified quantitative trait loci (QTL) and candidate genes associated with oil content in canola using linkage mapping or genome-wide association studies (GWAS) [39,40,41,42]. Xiao et al. [43] provided valuable insights into the genetic basis of seed oil content (SOC) in *B*. *napus* and identified loci and candidate genes that can be utilized in molecular breeding strategies to increase SOC in this important seed crop. That study identified 1713 candidate genes in the 26 QTLs confidence interval. To narrow down the focus, differentially expressed genes (DEGs) were identified between high- and low-SOC accessions, leading to the identification of 21 candidate genes related to SOC. Another study collected 204 accessions and recombinant inbred lines (RILs) from a natural population for genome-wide association analysis (GWAS) and QTL mapping [44]. Based on whole-genome re-sequencing, a high-density linkage map was constructed, and a novel locus qA07.SOC was identified as the major QTL for SOC based on the GWAS and RIL populations. In the qA07.SOC interval, the candidate genes BnaA07.STR18, BnaA07.NRT1, and BnaA07g12880D were predicted.

Fatty acid composition is another important quality trait in canola, as it affects the nutritional and industrial properties of the oil. Canola oil contains mainly unsaturated fatty acids, such as oleic acid (C18:1), linoleic acid (C18:2), and linolenic acid (C18:3), which have beneficial effects on human health [45]. However, some applications of canola oil require specific fatty acid profiles, such as high oleic or low linolenic acid contents, to improve the stability and functionality of the oil [46,47]. Several studies have identified QTL and candidate genes associated with fatty acid composition in canola using linkage mapping or GWAS [48,49,50]. For example, using a panel of 300 inbred lines, Zhu et al. [50] employed genome-wide association analysis to discover significant single-nucleotide polymorphism (SNP) sites linked with four fatty acid content parameters in rapeseed. A total of 148 SNP loci were discovered, including 30 loci on the A08 and C03 chromosomes that were linked to three characteristics at the same time. The researchers also identified 108 extremely advantageous genes for raising oleic and linoleic acid concentration while decreasing erucic acid levels. Using BLAST, 20 orthologs of functional candidate genes related to fatty acid biosynthesis were found within 100 Kb of significant SNP-trait relationships.

Protein content is another important quality trait in canola, as it determines the value and quality of canola meals for animal feed. Protein content is moderately heritable and varies from 17 to 26% among canola varieties [51]. Several studies have identified QTL and candidate genes associated with protein content in canola using linkage mapping or GWAS [52]. However, the genetic basis of protein content in canola is still not fully understood, and the QTL detected so far explains only a small proportion of the phenotypic variation. Therefore, more comprehensive and fine-scale QTL mapping and candidate gene analyses are needed to reveal the molecular mechanisms underlying protein content in canola and facilitate marker-assisted selection for this trait. Gazave et al. [53] evaluated a diverse panel of spring-type *B*. *napus* accessions for phenotypic variations in seed fatty acid content and composition at four US locations over multiple years. They observed highly heritable variations for total oil content, nine fatty acid compounds, and fourteen derivative traits. A GWAS detected 53 SNPs significantly associated with one or more of the twenty-four fatty acid seed traits, implicating the genetic role of twelve candidate genes, including four with two homologs each from the acyl-lipid pathway. Whole-genome prediction showed moderately high predictive abilities (70% of traits with abilities > 0.70), suggesting that these traits are highly amenable to genomic selection.

Glucosinolates are sulfur-containing compounds that have anti-nutritional effects on animals and humans, such as reducing iodine uptake, impairing thyroid function, and causing goiter [54]. Therefore, one major objective for improving the value and safety of canola meals has been breeding for low GSL content in canola seeds [55]. Breeding for low seed GSL content has reduced the negative effects of GSLs on animal feed and decreased the plant’s resistance to disease and insects [56,57]. Therefore, there is a need to identify genes that control GSL transport from vegetative tissues to seeds and to create new genotypes of canola with low seed GSL content but high disease resistance [58].

Recently, a GWAS of six GSL metabolites was conducted in rapeseed, including three aliphatic GSLs (m145 gluconapin, m150 glucobrassicanapin, and m151 progoitrin), one aromatic GSL (m157 gluconasturtiin), and two indole GSLs (m165 indolylmethyl GSL and m172 4-hydroxyglucobrassicin) [59]. There were 113 potential intervals discovered, all of which were closely related to these six GSLs metabolites. Based on the GWAS and analysis of differentially expressed genes, 187 candidate genes involved in GSL biosynthesis (e.g., BnaMAM1, BnaGGP1, BnaSUR1, and BnaMYB51) and novel genes (e.g., BnaMYB44, BnaERF025, BnaE2FC, BnaNAC102, and BnaDREB1D) were predicted. This study sheds light on the genetic basis of GSL biosynthesis in rapeseed, aiding marker-based breeding for better seed quality in Brassica species. To enhance pest and disease resistance in *B*. *napus*, it’s important to increase leaf GSL content while keeping seed content low. Liu et al. [56] found a strong correlation (r = 0.79) between seed and leaf GSL content in their study of 366 accessions. They identified seventy-eight loci associated with GSL traits, including five common and eleven tissue-specific loci related to total leaf and seed GSL content, and inferred thirty-six candidate genes involved in GSL biosynthesis. They also validated the candidate gene BnaA03g40190D (BnaA3.MYB28) responsible for high leaf/low seed GSL content. These findings shed light on the genetic basis of GSL variation and could facilitate metabolic engineering and breeding for high leaf/low seed GSL content in *B*. *napus*.

Agronomic traits in canola, including yield potential, disease resistance, herbicide tolerance, and shatter resistance, are complex and influenced by both genetics and environmental factors, such as drought and plant density [60]. Recent advances in genomics and transcriptomics have enabled the identification of QTLs and candidate genes associated with these traits. In a recent study, the genomic basis for the selection of adaptation and agronomic traits in rapeseed breeding was assessed through whole-genome resequencing of 418 diverse rapeseed accessions [39]. A total of 628 loci associated with causative candidate genes for 56 agronomically important traits, including plant architecture and yield traits, were identified through genome-wide association studies. The study also revealed nonsynonymous mutations in candidate genes with significant differences in allele frequency distributions across the improvement process, such as the BnRRF gene for seed weight.

Menendez et al. [61] used 152 accessions and a double-haploid population of 99 lines to investigate how plant density impacts *B*. *napus* growth and yield. Significant relationships have been identified for twelve traits at two densities, and transcriptome analysis demonstrated gene-regulatory responses to simulated shade. Candidate genes were identified within canola QTLs, and their effects on density responses in Arabidopsis lines were investigated. Within canola QTLs, candidate genes were identified, and their impacts on density responses in Arabidopsis lines were studied. TCP1 gene supports growth regardless of density, but HY5 especially enhances biomass and yield at high density. PIN genes accelerate blooming time in a density-dependent manner, whereas FT, HY5, and TCP1 act in a density-independent manner. Combining agronomic field experiments with genetic and transcriptomic approaches can improve crop productivity.

### 2.2. Environmental Factors

During the crop cycle, the quality of canola grains is significantly influenced by weather conditions such as air temperature, rainfall, relative humidity, and photoperiod, especially during the flowering and seed-filling stages [62,63]. Temperature affects the oil content and fatty acid proportion of canola grains by regulating the activity and expression of enzymes involved in lipid biosynthesis [64]. High temperatures during the flowering stage can cause reduced seed set and lower oil content, while excessive heat during seed maturation can lead to increased green seed, altered fatty acid composition, and reduced oil quality in canola grains [64,65]. In contrast, lower temperatures can delay flowering and result in reduced yield, and during the vegetative stage, they can also affect canola growth and yield by causing frost damage, reduced seed size, increased moisture content, reduced oil content, altered fatty acid composition, and increased sinapine content [12,66,67]. During the reproductive phase, both very low and very high temperatures may result in flower abortion and the fall of seedpods, causing greater unevenness in the maturation of the crop and interfering with the efficiency and quality of the oil produced [68].

Precipitation and soil moisture availability are crucial factors for canola production because water stress can impact canola grain yield and quality by influencing various physiological processes, including photosynthesis, respiration, transpiration, nutrient uptake, and metabolism [69]. Drought stress can decrease canola grain yield and quality by causing oxidative stress and lipid peroxidation in canola grains, which results in higher FFA content, decreased oil content, altered fatty acid composition, increased GSL content, and reduced protein content [4,70]. Sinaki et al. [71] discovered that irrigating at 50% depletion of available soil water increased grain yield, oil content, and protein content compared to irrigating at 75% depletion of available soil water or rainfed conditions. Water stress can also affect the physical properties of canola grains, such as seed size, shape, color, density, hardness, and test weight [71]. Proper water supply during the vegetative stage can lead to better plant growth and yield, while excessive water can result in diseases that have an adverse impact on grain quality [72].

Soil nutrients, particularly nitrogen (N), phosphorus (P), potassium (K), and sulfur (S), are essential for canola growth and grain quality [73,74]. Adequate soil fertility can lead to better plant growth and yield and higher oil content [75,76]. Adequate soil fertility can lead to better plant growth and yield and higher oil content [77,78]. However, excessive use of fertilizers can lead to the accumulation of nitrates in *B*. *napus* seeds, which can be harmful to human health [79]. Therefore, it is important to choose appropriate fertilization strategies and cultivars adapted to local soil fertility conditions.

Pests and diseases can affect the quality of canola grain by causing physical damage, nutrient loss, toxin accumulation, hormonal imbalance in plants, and reduced yield [80,81]. In addition, they can cause losses in oil content and quality, increase GSL content, reduce protein content and quality, and affect seed germination and vigor (Canola Council of Canada, 2020). Some of the major pests and diseases of *B*. *napus* are flea beetles, cutworms, diamondback moths, cabbage seedpod weevils, root maggots, *Sclerotinia* stem rot, blackleg, clubroot, and *Alternaria* black spot [82,83]. However, the diversity of species and the degree of infestation can vary widely between different countries. Therefore, effective pest and disease management practices are crucial for maintaining canola grain quality and ensuring profitable production.

### 2.3. Agronomic Practices and Management Techniques

Agronomic practices, including seed selection, sowing date, plant density, fertilization, irrigation, disease and pest control, and harvesting, influence canola grain quality [61,77,84,85,86]. They can impact the yield, oil content, fatty acid composition, protein content, GSL content, and chlorophyll content of canola grains [85,87,88,89,90]. Cultivars differ in oil content, protein levels, disease resistance, and other traits that can affect seed quality [86]; thus, choosing a canola cultivar suitable for the growing region is critical. In addition, optimal seeding rates and row spacing can promote uniform plant distribution, reduce competition for resources, and improve yield and grain quality [86,91,92].

The sowing date of canola affects the exposure of the crop to different weather conditions during its growth cycle, which may have an impact on grain yield and quality [91]. Canola should be planted at the optimal time for the local climate to maximize seed yield and quality [85]. For example, early seeding can improve canola emergence, reduce the risk of heat stress during flowering and seed filling, and increase the oil content and fatty acid composition and yield of canola grains [93,94]. In a study conducted by Righi et al. [95] in Brazil, there was a trend of reduction in grain productivity and quality with delayed sowing in four hybrids of canola sown on different dates. The authors found that one hybrid had better productivity and presented the best quality indicators such as oil content, protein content, chlorophyll content, GSL content, and fatty acid composition. The negative effect of delayed sowing was attributed to higher air temperatures during the filling-maturity sub-period and throughout the cycle.

Irrigation can affect the soil moisture status and water stress level of canola plants, which can influence the physiological processes and biochemical reactions of canola grains [9,96]. Especially under drought-prone conditions, it can improve canola grain quality by maintaining adequate soil moisture levels, which allows for optimal seed development [97]. However, excessive irrigation can lead to a reduction in oil content and an increase in undesirable fatty acids [98]. In addition to water uptake, adequate N and S nutrition can enhance the oil and protein content and fatty acid composition of canola seeds, while balanced P and K levels can improve seed quality [74,75]. Furthermore, the application of micronutrients, such as boron and zinc, may also improve grain quality, and their lack of nutrients reduces yield and seed size [99]. Fertilizer rate and placement can influence the nutrient uptake, biomass production, and oil quality of canola [100]. Side-banding fertilizer, for example, can reduce nitrogen losses and increase nitrogen use efficiency compared to broadcast application [100].

Crop rotation with different families of plants may help in breaking disease cycles and improving soil fertility [87]. Seed yield and use of fertilizer were studied for six years in Alberta, Canada, in twelve treatments, including continuous cropping and rotations of canola, wheat, pea, barley, and flax [101]. Canola yield increased with 1- or 2-year breaks from canola. Rotations over continuous canola increased canola yield by 0.632 Mg ha^−1^ (19.4%) on average from 2010 to 2015. Furthermore, nitrogen saving was observed when the pea plant was included in the rotation. Another study showed that green manure in *B*. *napus* rotation increased grain yield and nutrient uptake, while co-application of farmyard manure, compost, and chemical fertilizers increased grain yield and oil yield [102].

Weed control is essential for optimizing canola quality, as weeds can compete with canola for resources, reduce crop yield and quality, and increase dockage and green seed levels [92]. Timely weed removal with herbicides or mechanical methods can improve canola emergence, growth, yield, and quality [92]. However, overreliance on pre- and post-emergent herbicides has led to herbicide resistance in weeds [103]. Fortunately, herbicide-tolerant canola cultivars offer flexibility in weed management but increase the risks of outcrossing and weed shifts to resistance [104]. Integrated weed management techniques, including non-chemical options such as canola competition and allelopathy, are being examined to address these risks [105]. However, the effectiveness of non-chemical options in commercial fields is less proven than that of chemical herbicides. On-farm demonstrations and long-term research may improve the acceptance and use of these tactics [104].

Effective pest management strategies, including the use of resistant cultivars, crop rotation, and timely application of pesticides, can help minimize the damage caused by pests and diseases while maintaining high-quality canola grains [106,107]. On the other hand, insect pollination of canola is another crucial factor in increasing its seed yield and quality [108]. Honey bees are the most effective pollinators for canola, as they enhance both self-pollination and cross-pollination. Introducing at least three bee hives per hectare at the beginning of the flowering period is recommended to ensure a uniform and timely pod setting, which prevents losses and improves harvesting efficiency [109,110]. Therefore, managed pollinators are a valuable resource for canola growers and should be managed properly to maximize their benefits.

### 2.4. Harvest and Post-Harvest Management


*Harvesting, Drying, Storage and Transportation*


The quality of canola grains is influenced by post-harvest factors such as harvesting, drying, storage, and transportation. These factors can affect the physical and chemical properties and stability of canola grains and their derived products and can impact the nutritional and functional properties of the oil [111]. There are different methods of harvesting canola, such as direct combining, swathing, pushing, and desiccation, each with its own advantages and disadvantages depending on the crop and environmental conditions. Canola is ready to harvest when the seed moisture content is between 8 and 10% (dry matter), and the seed color has changed by 60% on the main stem [112]. The choice of harvest method depends on various factors such as crop maturity, uniformity, density, lodging, weather conditions, and risk of shattering. Hence, these methods can affect the yield and quality of canola grains by influencing seed loss, seed damage, moisture content, oil content, FFA content, GSL content, chlorophyll content, and germination capacity [113].

Harvesting too early or too late can affect the potential of canola seeds to develop secondary dormancy, which is a mechanism of seedbank persistence that can cause volunteer weed problems in subsequent crops. Secondary dormancy is influenced by both genetic and environmental factors in the mother plant. Canola seeds had a lower potential for secondary dormancy at early development but a higher potential at full maturity [114]. Therefore, harvesting canola at the optimal stage may reduce the ability of the seeds to develop secondary dormancy and thus reduce the persistence of seeds in the soil seed bank. Early harvesting can also result in immature grains, which can lead to lower oil content, higher levels of chlorophyll, and higher levels of FFAs in the oil, while late harvesting, on the other hand, can result in over-mature grains, which can lead to lower oil content and higher levels of impurities such as dirt and weed seeds [27]. Some canola cultivars have a high potential for seedpod shattering during harvest, resulting in yield loss [66]. In addition, a delayed harvest can result in an increase in pod shatter and reductions in seed yield. Thus, proper timing of harvest can maximize oil content and minimize losses due to shattering and weathering [115].

Drying is the process of reducing the moisture content of canola grains to a safe level for storage and processing. Drying methods include natural drying and artificial drying. Natural drying is the process of allowing the grains to dry naturally by air circulation in bins or silos. Artificial drying is the process of applying heat and forced air to the grains in dryers. Drying methods can affect the quality of canola grains by influencing the moisture content, oil content, FFA content, GSL content, chlorophyll content, and germination capacity [116]. Grain with a moisture level of 23% can be securely stored at 35 °C for fewer than 20 h. When relative humidity is constant, the grain’s equilibrium moisture content decreases by about 0.5% for every 10 °C increase in air temperature. When the moisture content of the grain is lowered to 14%, and the batch temperature is reduced to 25 °C, it can be securely stored for 35 days [117,118]. Adequate drying and storage conditions are essential for preventing spoilage and maintaining grain quality.

Cleaning and grading are essential post-harvest processes that remove impurities and separate canola grains according to size and density. Proper cleaning and grading can improve the quality of canola grains by reducing impurities and improving the uniformity of the grains. However, improper cleaning and grading can damage the grains and reduce their quality [119]. Canola grains are processed to extract oil, which can affect the nutritional and functional properties of the oil. The processing methods used, including solvent extraction, expeller pressing, and cold pressing, can impact the oil yield, color, flavor, and fatty acid composition [120]. The use of harsh processing methods can lead to oxidation and degradation of the oil, reducing its quality [121].

Post-harvest management, including storage and processing, plays a crucial role in maintaining canola grain quality [120]. Proper storage conditions (temperature, humidity, and aeration) are essential to prevent lipid oxidation, fungal growth, and mycotoxin production [122]. Additionally, the processing of canola seeds into oil and meal requires careful consideration of factors such as temperature, pressure, and solvent selection to retain the desired quality attributes [123,124]. Storage conditions (i.e., temperature, moisture, oxygen, insects, fungi, and rodents) can affect the quality of canola grains by influencing the oil stability, oxidative rancidity, FFA content, GSL content, chlorophyll content, and germination capacity [111,125,126]. Canola grains are susceptible to insect infestations and fungal growth during storage. High moisture content in the grains can promote fungal growth, while high temperature and humidity can lead to insect infestations [127]. Hence, proper storage conditions, including low temperature, low humidity, and adequate ventilation, can prevent these problems and preserve the quality of canola grains. Moreover, exposure to high temperatures and humidity during transportation can promote fungal growth and reduce the shelf life of grains [122].


*Quality Assessment Methods and Standards*


Quality assessment methods and standards for canola grains include various physical, chemical, and biological methods to evaluate the quality attributes such as moisture, test weight, oil content, protein content, fatty acid composition, GSL content, impurities, seed contaminants, and genetic purity [128]. These methods and standards ensure the safety, functionality, and marketability of canola products. Near-infrared spectroscopy, gas chromatography, and nuclear magnetic resonance are used to evaluate oil content, fatty acid composition, and other quality parameters. Grading standards classify canola grains based on factors such as moisture content, foreign material, and damaged seeds [119]. Some of these standards are presented in Table 3.

## 3. Implications for Canola Grain-Based Foods

The quality of canola grain is affected by several factors that occur during post-harvest processing and storage, including drying methods, storage conditions, milling methods, extraction methods, and refining methods [132]. Grain quality directly impacts the nutritional value, functional properties, and sensory attributes of canola-based foods such as oil, meal, flour, and bakery products [133,134]. Therefore, factors affecting canola grain quality have significant implications for the nutritional value, functionality, and safety of these foods [132].

### 3.1. Canola Oil


*Factors Affecting Canola Oil Extraction*


High-quality canola grains are essential for producing high-quality canola oil with a desirable fatty acid profile, low levels of impurities, and good oxidative stability [135]. However, the extraction efficiency of canola oil from canola seeds is not always optimal and can vary depending on several factors. These factors include seed pretreatment, moisture content, and extraction method [124]. Seed pretreatment corresponds to the process of preparing the seeds for oil extraction by removing impurities and hulls and reducing seed size. This process can improve the oil extraction efficiency by increasing the surface area of the seeds, reducing oil viscosity, and enhancing oil release [136]. However, excessive seed size reduction may have a negative impact on the oil yield and quality by causing oil loss, oxidation, or degradation. The optimal seed size reduction may depend on the type of extraction method and the seed moisture content [137].

Moisture content is an important factor that affects oil extraction efficiency. It influences the physical and chemical properties of both the seeds and the oil, such as density, viscosity, solubility, and stability [128]. Different extraction methods require different optimal moisture levels for canola seeds. For mechanical pressing, the optimal moisture content is between 6 and 10%, while for solvent extraction, it is between 3 and 5% [138]. Moisture content outside these ranges can lower the oil yield and quality.

The extraction method is another factor that influences oil extraction efficiency. The choice of the canola oil extraction method depends on several factors, such as oil yield, oil quality, energy consumption, capital cost, regulatory constraints, and environmental impact [132]. There are three main methods of extracting oil from canola seeds: mechanical pressing, solvent extraction, and supercritical fluid extraction [124]. Mechanical pressing is a physical method that uses pressure to squeeze out the oil from the seeds. Solvent extraction is a chemical method that uses a solvent, usually hexane, to dissolve and separate the oil from the seeds. Supercritical fluid extraction is a novel method that uses a fluid, usually carbon dioxide, at high pressure and temperature to extract the oil from the seeds [124]. Each method has its own advantages and disadvantages in terms of oil yield, quality, cost, and environmental impact. Mechanical pressing is more environmentally friendly and preserves more natural antioxidants in the oil, but it has a lower oil yield and more residual oil in the cake than solvent extraction [139]. Solvent extraction has a higher oil yield and lower residual oil in the cake than mechanical pressing, but it requires more energy and produces more waste than mechanical pressing. Supercritical fluid extraction has advantages over conventional solvent extraction methods, such as higher oil yield, higher diffusivity, lower viscosity and surface tension, and faster extraction times, but it is more expensive and complex than either mechanical pressing or solvent extraction [140].


*Factors Affecting Canola Oil Refining*


Canola oil needs to undergo several refining processes to improve its quality and stability before it can be used for human consumption or industrial applications. These processes include degumming, neutralization, bleaching, and deodorization. Crude oil quality affects the refining requirements and performance by influencing the amount and type of impurities present in the oil, such as phospholipids, pigments, FFAs, peroxides, and volatile compounds [141]. The quality of the crude oil may vary depending on the extraction method, storage conditions, and seed quality [137]. The higher the impurity content in the crude oil, the more intensive and costly the refining process will be.

Degumming is the first step in canola oil refining and aims to remove phospholipids and other impurities from crude oil. Phospholipids are undesirable in the oil because they can cause emulsification, foaming, and darkening during subsequent refining steps [142]. Degumming can be performed by adding water or acid to the oil and then separating the gums by centrifugation or filtration. The efficiency of degumming can be influenced by factors such as the amount of water or acid added and the temperature of the process. According to Gaber et al. [138], increasing the amount of water or acid can increase the removal of phospholipids but may also increase the loss of oil and antioxidants. Similarly, increasing the temperature can improve degumming efficiency but may also cause thermal degradation of the oil and antioxidants.

Neutralization is the second step of canola oil refining and aims to remove FFAs from the oil. FFAs are undesirable in the oil because they can cause rancidity, off-flavors, and a reduced smoke point [124]. Neutralization can be performed by adding an alkali solution, usually sodium hydroxide or sodium carbonate, to the oil and then separating the soap stock by centrifugation or filtration [142]. The efficiency of neutralization can be influenced by factors such as the concentration of the alkali solution and the temperature of the process [143]. Increasing the concentration of the alkali solution can increase the removal of FFAs but may also increase the loss of oil and antioxidants. Increasing the temperature, on the other hand, can boost neutralization efficiency while simultaneously causing thermal deterioration of the oil and antioxidants [143].

Bleaching is the third step in canola oil refining and aims to remove color pigments and other impurities from the oil. Color pigments are undesirable in the oil because they can affect its appearance and stability [144]. Bleaching can be performed by adding adsorption bleaching clay, activated carbon, or special silica to the oil, which adsorbs color pigments and other impurities [145]. The efficiency of bleaching can be influenced by factors such as the type and amount of bleaching earth used and the temperature and pressure of the process [145]. Increasing the amount of bleaching earth used can improve the removal of color pigments and other impurities, but it can also result in a loss of oil and antioxidants. Similarly, raising the temperature and pressure can boost bleaching efficiency while simultaneously causing thermal deterioration of the oil and antioxidants [146].

Deodorization is the final step of canola oil refining, which aims to remove odorous compounds from the oil. Odorous compounds are undesirable in the oil because they can affect its flavor and shelf life [143]. Deodorization can be performed by heating the oil under vacuum and passing steam through it. The steam strips off the odorous compounds and carries them away from the oil. The efficiency of deodorization can be influenced by factors such as temperature, pressure, and duration of the process. According to Cai et al. [121], increasing the temperature and pressure can increase the removal of odorous compounds, but it can also cause the thermal degradation of the oil and its antioxidants.

Refining conditions affect the quality and quantity of canola oil by influencing the extent and rate of impurity removal during degumming, bleaching, and deodorization processes [147]. Some refining conditions that may affect canola oil quality and quantity are temperature, pressure, time, water or acid concentration, bleaching earth or activated carbon dosage, steam flow rate, and vacuum level [143]. The optimal refining conditions may vary depending on the crude oil quality, refining method, and product specifications [141,147]. In any case, the refining process may alter the quality of the oil, eliminating undesirable compounds that alter the oil’s storability but also reducing the quantities of micronutrients and antioxidants [148,149].

### 3.2. Canola Meal

The quality and nutritional value of canola meal can vary depending on several factors [19]. These factors include the extraction method, protein content, and the presence of anti-nutritional factors. The extraction method used to obtain canola oil from canola seeds can influence the protein content and overall quality of the canola meal. In fact, the choice of extraction method can affect the protein content and quality of canola meal. According to Khajali et Slominski [18], solvent extraction typically results in a higher protein content (38–40%) compared to mechanical pressing (34–36%). However, solvent extraction can also cause more damage to the protein structure and reduce its digestibility and amino acid availability. Moreover, solvent extraction can leave residual hexane in the meal, which can pose health risks for humans and animals.

Desolventized flakes are heated during toasting to lower moisture content and enhance flavor and meal stability [150]. The quantity and quality of canola meals generated via desolventization and toasting can, however, be impacted by a number of factors. Temperature, time, pressure, as well as the composition and characteristics of the oilseed flakes are some of these factors. One of the most crucial elements that influence the quantity and quality of canola meal is temperature. It determines the rate and extent of solvent removal, as well as the degree of protein denaturation and Maillard reaction [18]. The type and quality of the oilseed flakes, as well as the required qualities of the meal, determine the proper temperature for desolventization and toasting. According to Daun et al. [150], desolventization and toasting often occur at temperatures between 95 and 115 °C. Higher temperatures have the potential to degrade both the yield and the quality of canola meals by causing more damage to the protein structure, reducing the availability of amino acids, and increasing the loss of oil and moisture from the flakes [150]. Moreover, the optimal pressure for desolventization and toasting depends on temperature, time, and the composition and characteristics of the oilseed flakes. The typical pressure range is between 0.5 and 1.5 bar, and the typical time range is between 30 and 60 min [150]. Cooking for longer periods can improve the solubility and digestibility of the proteins in canola meals by reducing the levels of anti-nutritional factors such as GSLs and tannins [17]. However, longer times and lower pressures can cause more damage to the protein structure, reduce its amino acid availability, and increase the loss of oil and moisture from the flakes, resulting in a lower yield and quality of canola meal [18,150].

Anti-nutritional factors are compounds that can interfere with the digestion and absorption of nutrients in animals. Canola meal contains several anti-nutritional factors, such as GSLs, phytic acid, and tannins [17]. GSLs are sulfur-containing compounds that can degrade into toxic metabolites, such as thiocyanates and goitrin. These metabolites can impair thyroid function, reduce iodine uptake, and cause goiter in animals [151]. Phytic acid is a phosphorus-containing compound that can bind to minerals such as calcium, iron, zinc, and magnesium. This can reduce the bioavailability of these minerals and cause mineral deficiencies in animals [152]. Tannins are polyphenolic compounds that can form complexes with proteins and carbohydrates. This can reduce the digestibility and utilization of these nutrients in animals [18]. The presence of these anti-nutritional factors can limit the use of canola meals as animal feed. However, selective breeding and processing techniques can help reduce the presence of these compounds. For example, low-GSL varieties of canola have been developed to improve the safety and palatability of canola meals for animals [153]. Moreover, processing techniques, such as heat treatment, enzymatic hydrolysis, and fermentation, can help reduce the levels of phytic acid and tannins in canola meal and improve its nutritional value [154].

### 3.3. Canola Flour and Bakery Products


*Factors Affecting Flour Production*


The production of canola flour from canola bran involves milling and sieving. Milling is a process that reduces the particle size and distribution of canola bran by using roller mills or hammer mills. Sieving is a process that separates fine particles from coarse particles using sieves or classifiers. The choice of production method depends on several factors, such as flour quality, product specifications, energy consumption, capital cost, and environmental impact [155]. Milling has the advantage of producing finer and more uniform particles of flour, which may improve its functional properties and end-use applications. However, milling has the disadvantage of consuming energy and producing heat, which may affect its nutritional value and stability [156]. While sieving has the advantage of producing different grades of canola flour, which may suit different product specifications and consumer preferences. However, sieving has the disadvantage of producing waste, which may affect its profitability and sustainability [157].

Particle size is one of the factors that affect the functional properties of canola flour. Particle size refers to the diameter of the canola flour particles, which can range from fine to coarse. Particle size can influence the water absorption capacity, oil absorption capacity, and pasting properties of canola flour [158]. Water absorption capacity is the ability of canola flour to retain water, which affects its hydration and swelling behavior. Oil absorption capacity is the ability of canola flour to retain oil, which affects its emulsification and dispersion behavior. Pasting properties are the changes in viscosity and consistency of canola flour when heated in water, which affects its gelatinization and retrogradation behavior. According to Ratnayake et al. [158], finer particle size (50 to 500 µm) can increase the water absorption capacity, oil absorption capacity, and pasting properties of canola flour compared to coarser particle size (>500 µm). However, finer particle size can also increase the dustiness and caking tendency of canola flour compared to coarser particle size [159]. Therefore, it is important to find the optimal particle size that maximizes the functional properties of canola flour without compromising its handling and storage characteristics.

Moisture content is another factor that affects the shelf life and stability of canola flour. Moisture content refers to the amount of water present in canola flour, which can vary depending on the drying conditions and storage conditions. Moisture content can affect the microbial growth, lipid oxidation, and enzymatic activity of canola flour [159]. Microbial growth is the proliferation of microorganisms, such as bacteria and fungi, in canola flour, which can cause spoilage and contamination. Lipid oxidation is the degradation of lipids in canola flour by oxygen, which can cause rancidity and off-flavors. Enzymatic activity is the catalysis of chemical reactions in canola flour by enzymes, such as lipases and proteases, which can cause hydrolysis and degradation of lipids and proteins. Lower moisture content can reduce the microbial growth, lipid oxidation, and enzymatic activity of canola flour compared to higher moisture content [159]. However, lower moisture content can also increase the brittleness and breakage of canola flour particles compared to higher moisture content [159]. Therefore, it is important to find the optimal moisture content that maximizes the shelf life and stability of canola flour without compromising its physical integrity.

Bran temperature is another factor that affects the milling and sieving efficiency of canola flour. Bran temperature refers to the temperature of canola bran during milling and sieving, which can vary depending on the heating conditions and ambient conditions. Bran temperature can affect the physical and chemical properties of canola bran, such as hardness, elasticity, brittleness, and solubility [159]. These properties affect the ease and extent of bran size reduction and separation during milling and sieving. According to Aider and Barbana [159], higher bran temperatures can increase the milling and sieving efficiency of canola flour compared to lower bran temperatures. However, higher bran temperatures can also cause more damage to the protein structure and reduce its amino acid availability [18]. Moreover, higher bran temperatures can also increase the loss of oil and moisture from the bran, resulting in a lower yield and quality of canola flour [159]. Therefore, it is important to find the optimal bran temperature that maximizes the milling and sieving efficiency of canola flour without compromising its protein quality and quantity.


*Factors Affecting Bakery Products*


The use of canola oil, meal, or flour in bakery products involves several steps, such as mixing, kneading, proofing, baking, and cooling. The choice of bakery product formulation and processing method depends on several factors, such as product quality, product specifications, consumer preferences, cost, and availability [160]. Canola oil, meal, or flour may be used in bakery products to replace or supplement other ingredients such as wheat flour, fat, or eggs. Canola oil may be used to provide moisture, tenderness, flavor, and shelf-life to bakery products [161,162]. Canola meal may be used to provide protein, fiber, minerals, and antioxidants to bakery products [120].

Ingredient composition affects the quality and quantity of bakery products by influencing the proportion and balance of different components in the bakery product, such as flour, fat, sugar, water, eggs, leavening agents, and additives [163]. This composition may vary depending on the type and purpose of the bakery product (e.g., bread, cake, muffin, cookie, or pastry) and the level and type of canola oil, meal, or flour used in the bakery product. The optimal ingredient composition depends on the desired sensory and nutritional attributes of the product [163]. On the other hand, ingredient quality can influence the physical and chemical properties of different components in the bakery product, such as moisture content, oil content, protein content, fiber content, gluten content, starch content, sugar content, FFA content, chlorophyll content, GSL content, and antioxidant content [163,164]. It may vary depending on the source and processing of the ingredients, such as canola oil, corn meal, or flour. The ingredient quality may also vary depending on the storage and handling of the ingredients, such as temperature, humidity, light exposure, and oxygen exposure.

Ingredient interaction affects the compatibility and synergy of different components in the bakery product, such as flour-fat, flour-water, flour-protein, flour-starch, flour-sugar, fat-water, fat-protein, fat-starch, fat-sugar, water-protein, water-starch, water-sugar, protein-starch, protein-sugar, and starch-sugar interactions [165,166]. The ingredient interaction may vary depending on the type and level of canola oil, meal, or flour used in the bakery product, as well as the processing conditions such as mixing, kneading, proofing, baking, and cooling. Processing conditions affect the quality and quantity of bakery products by influencing the physical and chemical changes that occur during different stages of bakery product processing, such as mixing, kneading, proofing, baking, and cooling. Some of the processing conditions that may affect bakery products made with canola oil, meal, or flour are mixing time, mixing speed, mixing temperature, kneading time, kneading speed, proofing temperature, proofing humidity, baking time, baking temperature, baking humidity, and cooling time.

Storage conditions also have an impact on the physical and chemical changes that occur during different stages of bakery product storage, such as packaging, transportation, and distribution [167,168,169]. Some of the storage conditions that may affect bakery products made with canola oil, meal, or flour are storage time, storage temperature, storage humidity, storage light, and storage oxygen exposure. The optimal storage conditions for bakery products depend on the desired stability, freshness, safety, and quality of the product.

## 4. Challenges and Prospects

Canola quality is affected by various factors, such as genetic variation, environmental conditions, agronomic practices, harvesting methods, storage conditions, and processing techniques. Therefore, there is a need for continuous research and innovation to improve canola quality and meet the changing demands of consumers and industries. Some of the future research needs and challenges for improving canola quality and its implications for grain-based foods are discussed below.

Improving oil quality: One of the main objectives of canola breeding is to improve the oil quality by increasing the content of desirable fatty acids and reducing the content of undesirable ones. For example, increasing the oleic acid content can improve the oxidative stability and shelf life of canola oil, while reducing the linolenic acid content can reduce the need for hydrogenation and trans-fat formation. Moreover, increasing the content of omega-3 fatty acids can increase the nutritional value and health benefits of canola oil [131]. However, improving oil quality also poses some challenges, such as maintaining yield potential, agronomic performance, disease resistance, and adaptation to different environments. Furthermore, there is a need to develop reliable and rapid methods for measuring oil quality traits in breeding programs and quality assurance systems.

Improving protein quality: Another objective of canola breeding is to improve protein quality by increasing the digestibility, enhancing the amino acid profile, and reducing anti-nutritional factors such as GSLs. Improving protein quality can increase the value and utilization of canola meals for animal feed and human food applications. For example, canola meals can be used as a protein ingredient in grain-based foods such as breads, cakes, cookies, noodles, pasta, and snacks. However, improving protein quality also faces some challenges, such as maintaining or improving oil content and quality, overcoming genetic limitations and trade-offs, developing novel processing methods to enhance protein functionality and palatability, and ensuring food safety and regulatory compliance.

Improving chlorophyll content: Chlorophyll is a pigment that gives canola seeds their green color. It is an undesirable component in canola oil because it reduces its clarity and stability, increases its refining cost, and affects its sensory properties. Therefore, reducing chlorophyll content is another goal of canola breeding and processing. The chlorophyll content is influenced by genetic factors as well as environmental factors such as temperature, moisture stress, frost damage, insect damage, disease infection, premature harvest, and delayed swathing. Therefore, there is a need for developing cultivars with low chlorophyll content or high chlorophyll degradation capacity under various conditions. Moreover, there is a need to develop novel methods to reduce chlorophyll content during post-harvest and processing stages, such as using enzymes and antioxidants.

Improving end-use quality: Canola oil and meal can be used for various end-use applications, such as food, feed, fuel, and industrial products. However, different end-use applications may have different quality requirements and specifications. Therefore, there is a need to improve the end-use quality of canola products by optimizing the processing conditions, modifying the functional properties, enhancing the sensory attributes, and ensuring safety and quality standards. For example, canola oil can be used for frying, baking, salad dressing, margarine, shortening, and biodiesel production. However, each of these applications may require different oil characteristics, such as smoke point, oxidative stability, flavor, color, viscosity, and fatty acid composition. Similarly, canola meals can be used for making bread, cakes, cookies, noodles, pasta, snacks, and animal feed. However, each of these applications may require different meal characteristics, such as protein content and quality, water absorption capacity, oil binding capacity, emulsifying properties, gelation properties, and flavor.

Improving canola processing: The United Nations’ sustainable development goals require the oilseed processing industry to use solvent extraction processes that are safe and eco-friendly. This poses a challenge, as the industry needs to find and optimize alternative and greener solvents that can process oilseeds without compromising the oil quality [170]. Furthermore, the industry should adopt a sustainable processing method and system that can efficiently recover the oil fraction from the oilseeds while preserving its quality and removing unwanted compounds [171].

## 5. Conclusions

The quality of canola grains and their derived products, such as canola oil, meal, flour, and bakery products, is influenced by a myriad of factors throughout their production, post-harvest processing, and storage. These factors can impact the chemical composition, physical properties, functional characteristics, and sensory attributes of canola grains and their products, which in turn affect their suitability for various food applications. This literature review has delineated the major factors affecting canola quality, including genetic factors, agronomic practices, environmental conditions, post-harvest processing techniques, and storage conditions. It has also highlighted the importance of optimizing these factors to ensure the safety, stability, and nutritive value of canola grains and their derived products. Future research should focus on the development of improved canola varieties with enhanced quality traits and better resistance to biotic and abiotic stresses and the assessment of the environmental, economic, and social impacts of canola production and consumption. Additionally, the optimization of agronomic practices, post-harvest processing methods, and storage conditions is crucial in maintaining the quality of canola grains and their products. Innovative technologies and processing techniques should be explored to minimize the impact of detrimental factors on canola quality while preserving its favorable nutritional and functional properties. Canola production must also adapt to the changes in terms of climatic, agricultural, economic, and regulatory conditions that affect the industry. This requires further research and discussion on the adaptation levers that can be put forward in the future. A multidisciplinary approach involving scientists, researchers, industry professionals, and policymakers is essential in order to develop sustainable, efficient, and innovative solutions that will contribute to the improvement of canola grain quality and the overall success of the canola industry. By addressing these challenges and maximizing the potential of canola grains and their derived products, we can contribute to the development of healthier and more sustainable food systems that will benefit consumers, producers, and the environment.

## Figures and Tables

**Figure 1 foods-12-02219-f001:**
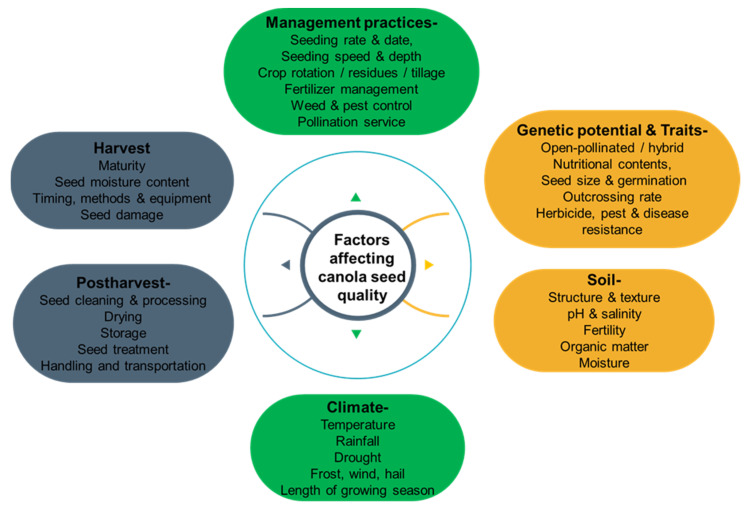
Factors affecting the quality of canola grains.

**Table 1 foods-12-02219-t001:** Quality data for 2022 and 2021 harvest samples and 5-year means for Canola grown in Western Canada *.

Quality Parameter	2022	2021	2017 to 2021 Mean
Oil content (%, 8.5% moisture)	42.1	41.3	43.8
Protein content (%, 8.5% moisture)	22.4	24.0	21.3
Oil-free protein of the meal (%, 12% moisture)	39.9	42.0	39.2
Chlorophyll content (mg/kg in seed)	9	10	11
Total seed glucosinolates (µmol/g, 8.5% moisture)	12	12	10
Oil-free total glucosinolates of the meal (µmol/g, 8.5% moisture)	21	20	19
Free fatty acids (%)	0.26	0.24	0.17
Oleic acid (% in oil)	64.6	64.2	63.5
α-Linolenic acid (% in oil)	8.2	8.6	9.2
Total saturated fatty acids (% in oil)	6.9	6.6	6.6
Iodine value (unit of oil)	109.5	110.9	112.0
Total mono-unsaturated fatty acids (% in oil)	66.1	65.8	65.1
Total poly-unsaturated fatty acids (% in oil)	26.3	27.0	27.7

* Data from the Canadian Grain Commission Harvest Surveys (http://www.grainscanada.gc.ca) (accessed on 22 May 2023).

**Table 2 foods-12-02219-t002:** Essential amino acid profile (g/100 g of dry matter) of canola meal compared with soybean meal, a plant reference protein, and the FAO/WHO-suggested requirements (adult) of essential amino acids [20,21].

Amino Acid	Canola Meal	Soybean Meal	FAO/WHO Requirements for Adults
Histidine	2.7	2.6	1.5
Isoleucine	4.0	4.0	3.0
Leucine	7.0	7.8	5.9
Lysine	5.8	6.4	4.5
Methionine	1.9	1.3	2.2
Phenylalanine	3.8	5.0	3.8
Threonine	4.5	4.0	2.3
Tryptophan	1.3	1.3	0.6
Valine	5.0	4.8	3.9

**Table 3 foods-12-02219-t003:** Quality assessment methods and standards for canola grains [129,130,131].

Quality Parameter	Definition	Measurement Method	Limit or Range
Moisture	Amount of water present in canola grains	Oven drying or electronic moisture meters	Maximum limit of 8.0%
Test weight	Weight per unit volume of canola grains (i.e., reflect the density, size, shape, and uniformity of seeds)	Standardized test weight apparatus	Minimum limit of 62 kg/hl
Oil content	Amount of oil present in canola grains	Nuclear magnetic resonance (NMR) or near-infrared reflectance (NIR) spectroscopy	No minimum or maximum limit
Protein content	Amount of protein present in canola grains	Combustion analysis or NIR spectroscopy	No minimum or maximum limit
Fatty acid composition	Proportion of different fatty acids present in canola oil	Gas chromatography (GC) or NIR spectroscopy	No specific limits, market preferences for certain fatty acid profiles may exist
Glucosinolate content	Amount of glucosinolates present in canola meal	High-performance liquid chromatography (HPLC) or NIR spectroscopy	Maximum limit of 30 μmol/g
Impurities	Foreign materials present in canola grains	Sieving and visual inspection methods	Maximum limit of 3.0% by weight
Seed contaminants	Weed seeds or other seeds present in canola grains (Seed contaminants classified into five types, A to E, based on their potential harm to humans, animals, or plants)	Counting methods per half L or per 2.5 L of canola grains	Maximum limits vary by type of contaminant
Genetic purity	Degree of conformity of canola grains to a specific variety or trait affects the identity preservation and traceability of canola products.	Molecular markers or bioassays	Different standards for conventional, GM (herbicide-tolerant), or specialty (high-oleic) varieties

## Data Availability

Data is contained within the article.

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
