# Peer review of "Factors Affecting the Quality of Canola Grains and Their Implications for Grain-Based Foods"

_foods, 2023, doi:10.3390/foods12112219_

Round 1
Reviewer 1 Report
I consider the review of the literature on the factors determining the quality of oilseed rape seeds and their suitability for food and feed purposes to be valuable. This study broadly covers the issue of rapeseed, so to speak, from "field to table". It is worth emphasizing that this review was based mostly on the latest scientific publications (166).
I suggest a few minor corrections and verifications.
In the introduction in line 41, I propose to change the order of the components according to importance: oil, protein ....water in the end. In this case water content will be better instead moisture.
On page 2 -The amino acid composition.
In seeds of rapeseed the content of sulfur amino acids is rather high, methionine and cystine content is higher than in soybean meal, but lysine and leucine is lower.
On page 4: the abbreviation SSR should be applay the first time you use the term "...84 simple sequence repeat markers (SSR)".
On page 5: literature references are missing in case Hu et al. 2022 and Liu et al. 2019.
On page 6 , line 256 is Temperature affects ....fatty acid composition but it should be fatty acid proportions.
On line 260 is High temperature increased....levels of erucic acid and GSL content in canola grains. However varieties canola type contain no erucic acid, maximum up to 2%.
Author Response
Dear Reviewer,
find in attached file our responses
Best regards

Reviewer 2 Report
Manuscript Title: Factors Affecting the Quality of Canola Grains and Their Implications for Grain-Based Foods
Manuscript ID: foods-2402171
This manuscript reviewed these affecting the quality of canola grains and their derived products. Meanwhile, the review also suggested future research needs and challenges for enhancing canola quality and its utilization in food. This paper has important guiding significance for the canola-based food processing. However, the article is too broad and not focused enough, I suggest that this article should be revised and re-reviewed.
Comments:
1. The content of this paper is too broad and not focused enough. For example, it is enough to write about the progress on the implications of canola grain-based foods, or canola oil processing.
2. In the introduction, it is suggested that the author should add some related data of planting area, yield, origin and varieties of canola.
3.It would be better if Table 1 provided data on the components of canola rather than the essential amino acids.
Author Response

(The authors gave the same response as above.)

Round 2
Reviewer 2 Report
The manuscript is now acceptable.